# Advancing Understanding of Lake-Watershed Hydrology: A Fully Coupled Numerical Model Illustrated by Qinghai Lake

Lele Shu[1,2,3,†], Xiaodong Li[4,†], Yan Chang[1,2,*], Xianhong Meng[1,2,3,*], Hao Chen[1,2,3,5], Yuan Qi[1], Hongwei Wang[1], Zhaoguo Li[1,2], and Shihua Lyu[6]

[1]Key Laboratory of Land Surface Process and Climate Change in Cold and Arid Regions, Chinese Academy of Sciences, Lanzhou, Gansu 730000, China
[2]Northwest Institute of Eco-Environment and Resources, Chinese Academy of Sciences, Lanzhou, Gansu 730000, China
[3]University of Chinese Academy of Sciences, Beijing 101408, China
[4]Qinghai Institute of Meteorological Sciences, Xining, Qinghai 810001, China
[5]College of Atmospheric Sciences, Lanzhou University, Lanzhou, Gansu 730000, China
[6]Chengdu University of Information Technology, Chengdu, Sichuan 610103, China
[†]These authors contributed equally to this work.

**Correspondence:** * Xianhong Meng (mxh@lzb.ac.cn) and Yan Chang (changy63@lzb.ac.cn)

**Abstract.** Understanding the intricate hydrological interactions between lakes and their surrounding watersheds is pivotal for advancing hydrological research, optimizing water resource management, and informing climate change mitigation strategies. Yet, these complex dynamics are often insufficiently captured in existing hydrological models, such as the bi-direction surface and subsurface flow. To bridge this gap, we introduce a novel lake-watershed coupled model, an enhancement of the Simulator

of Hydrological Unstructured Domains (SHUD). This high-resolution, distributed model employs unstructured triangles as its fundamental Hydrological Computing Units (HCUs), offering a physical approach to hydrological modelling. We validated our model using data from Qinghai Lake in China, spanning from 1979 to 2018. Remarkably, the model not only successfully simulated the streamflow of the Buha River, a key river within the Qinghai Lake Basin, achieving a Nash-Sutcliffe efficiency (NSE) of 0.62 and 0.76 for daily and monthly streamflow, respectively, but also accurately reproduced the decrease-increase

U-shaped curve of lake level change over the past 40 years, with an NSE of 0.71. Uniquely, our model distinguishes the contributions of various components to the lake's long-term water balance, including river runoff, surface direct runoff, lateral groundwater contribution, direct evaporation, and precipitation. This work underscores the potential of our coupled model as a powerful tool for understanding and predicting hydrological processes in lake basins, thereby contributing to more effective water resource management and climate change mitigation strategies.

## 1  Introduction

Lakes, as crucial components of the global hydrosphere, occupy only a small fraction of the earth's surface though, they play an indispensable role in the water cycle (Grant et al., 2021; Woolway et al., 2021; Pi et al., 2022). They are particularly susceptible to the impacts of climate change and anthropogenic activities, underscoring their significance in the global ecological balance. These water bodies serve as sensitive indicators of environmental changes, and their health and sustainability are of paramount

importance (Li et al., 2022; Jones et al., 2022; Woolway, 2023). The study of lake hydrology, therefore, is not only a scientific endeavor but also a necessity for effective environmental management and policy-making (Crowe and Schwartz, 1981; Carter, 1986; Qi et al., 2020).

The development and application of coupled lake-watershed models have emerged as a powerful approach to understanding the complex interactions between lakes and their surrounding landscapes. These models integrate various hydrological processes, including surface runoff, groundwater flow, and lake-water interactions, thereby providing a holistic understanding of the basin system. The use of such models has proven instrumental in studying regional water resources, nutrient loads in lakes and groundwater, hydrologic connectivity of lake systems, and the impact on local economies (Dong et al., 2019; Cobourn et al., 2018; Ladwig et al., 2021).

However, the development and application of these models are not without challenges. One of the key issues is the accurate representation of subsurface groundwater flow, which often has a more significant impact on lake water and chemical budgets than surface water inflow. Furthermore, the relative importance of surface water vs. groundwater, and watershed vs. in-lake processes needs to be accurately captured for effective lake understanding and management (Johnston and Shmagin, 2006).

Over the past few decades, the development of lake-watershed models has significantly advanced, driven by a growing recognition of their importance in understanding and managing aquatic ecosystems. These models have evolved from simple, lumped or semi-distributed models to more complex, spatially distributed models that can simulate a wide range of hydrological processes (Lewis et al., 1984; Kratz et al., 1997). Various researchers have further expanded upon this foundation. A noteworthy example is the WATLAC model, devised by Zhang and Werner (2009), which can emulate surface and subsurface fluxes into a lake, and fathom the interaction between a watershed and its lake (Ye et al., 2011). Similarly, the Soil and Water Assessment Tool and the Snowmelt Runoff Model have been used to simulate the water equilibrium within a lake's watershed (Zhang et al., 2014; Dargahi and Setegn, 2011). In a more recent work, one-way coupling was accomplished using the MGH-IPH hydrological model and the IPH-ECO hydrodynamic model, with application to Lake Mirim, in South America (Possa et al., 2022; Munar et al., 2018). Wu et al. (2017) integrated the hydrological model HIMS with a hydrodynamic model to examine water exchange in the Hongfeng Reservoirs. Xu et al. (2007) coupled the HSPF and CE-QUAL-W2 model to simulate the hydrodynamics and water quality in Lake Manassas and the Occoquan Reservoir in Virginia, USA. Chauvelon et al. (2003) combined the HIC and RMA2 model to replicate the Vaccares lagoon level and salinity, while Inoue et al. (2008) implemented a coupled hydrology-hydrodynamic model to simulate the hydrology and hydrodynamics in the Barataria Basin, USA. Dargahi and Setegn (2011) constructed the SWAT+GEMSS model to simulate Lake Tana, Ethiopia. These models have been instrumental in shedding light on the water cycle and runoff mechanisms in various lake basins, thereby illuminating the environmental issues plaguing these regions (Cui and Li, 2015a). Despite these advancements, further research is needed to improve the accuracy and reliability of these models, particularly in the challenges of lake-watershed bi-directional water exchange.

The conventional approach in these studies has been to leverage hydrological models to calculate the runoff to the lake, with the results then serving as boundary conditions in the hydrodynamic model. Such models typically consider the lake as an isolated water body, separate from the basin's water cycle, thus assuming that lake fluctuations do not influence watershed

groundwater or river discharge. In essence, most of these models operate on a one-way coupling scheme. However, a recent advancement by Ladwig et al. (2021) deviates from two-way coupling norm. They developed the PIHM-Lake model for bi-directional hydrological interactions between lake and watershed, and coupled it with a one-dimensional hydrodynamic-ecological model (GLM-AED2), which simulates the thermal and water quality dynamics of Lake Mendota in the USA. Uniquely, this model is capable of simulating both river runoff and fluxes in the lake's surrounding watershed.

Though a fully coupled lake-watershed hydrological model may not offer the highest resolution as typical of hydrodynamic models, it should accurately represent the bi-directional water exchange—encompassing surface, subsurface, and rivers—between the lake and its watershed. The combination of lake and hydrological models can simulate the lake's hydrology and surrounding landscape over time, thereby informing decisions geared towards the preservation of the lake's water quality and ecosystem health. This holistic understanding of bi-directional hydrological processes provides critical guidance for management decisions aimed at maintaining both the water quality and ecological health of the lake.

This study aims to develop and validate a fully coupled lake-watershed hydrological model, using Qinghai Lake in China as the test site. As the largest lake in China, Qinghai Lake's unique hydrological and environmental features make it an ideal candidate for this purpose. Its endorheic nature simplifies the lake-watershed connections, aiding in the model's testing. The extensive existing research data on Qinghai Lake's hydrology, ecology, and climate is instrumental for the model's calibration and validation. Additionally, the lake's characteristics as a high-altitude, cold, and arid region make it a representative model for similar environments, enhancing the model's interdisciplinary research potential and broadening its applicability.

The methodologies employed in the modeling process are thoroughly expounded upon in Section 2, while Section 3 showcases the simulation results for Qinghai Lake. Section 4 offers a discussion on the limitations of the model and potential areas for future enhancement.

## 2   Methods

### 2.1   Research Area

Qinghai Lake, the largest saline lake in China, is nestled within the Qinghai-Tibet Plateau in northwest China. Covering an expansive area of approximately 4,489 $\text{km}^2$ and plunging to a maximum depth of 32 m, this lake has emerged as a significant hydrological and environmental research hub. It has been the subject of extensive multidisciplinary research spanning hydrology, climate science, meteorology, limnology, ecology, and biogeochemistry (Zhang et al., 2014; Qi et al., 2020; Su et al., 2020; Wang et al., 2022).

The hydrological equilibrium of Qinghai Lake is primarily sustained by several rivers and streams, with no outlets. The Buha River is the most substantial contributors, responsible for ~49% of the total inflow into Qinghai Lake (Wang et al., 2022). The lake's water balance is intricately woven with precipitation, evaporation, and river discharge. As such, shifts in climate and land use patterns can profoundly influence this balance. The lake's climate, typified by an average annual temperature of approximately -2.5 °C and an average annual precipitation of 415 mm (based on CMFD reanalysis data, 1919-2018), is largely dictated by the Asian monsoon system, with the majority of rainfall occurring during the summer months. Climate alterations,

such as temperature escalations and precipitation pattern changes, can significantly affect the lake's water balance, ecology, and biogeochemistry (Su et al., 2019, 2020).

Qinghai Lake, a unique ecosystem with a high degree of endemism, provides a haven for a diverse array of endemic fish and bird species. Changes in the lake's water chemistry and hydrology can have substantial impacts on its ecology. Consequently, Qinghai Lake serves as a pivotal research site for exploring the complex interconnections between hydrology, climate, meteorology, limnology, ecology, and biogeochemistry in high-altitude, cold, and arid regions.

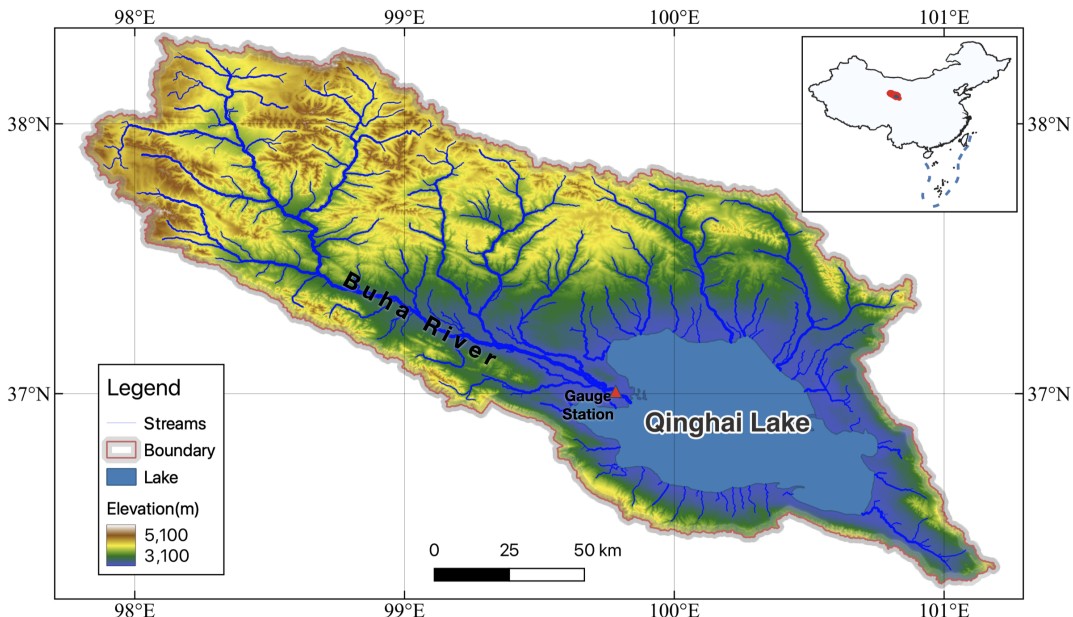

**Figure 1.** Research Area map with elevation, river network, and stream gauge stations.

## 2.2    SHUD Model

The Simulator for Hydrologic Unstructured Domains (SHUD) is a comprehensive, multi-scale, integrated surface-subsurface numerical hydrological model (ISSNHM) that employs the Finite Volume Method and unstructured triangular mesh. The merits of ISSNHMs lie in their temporal-spatial continuum, contrasting with other semi-distributed models like SWAT, TOP-MODEL, VIC etc. (Freeze and Harlan, 1969; Hrachowitz and Clark, 2017). This model, an evolution of the computational strategy developed by Qu and Duffy (2007) for the Penn State Integrated Hydrologic Model (PIHM), provides a more accurate
depiction of a watershed's physical attributes, such as topography and land use patterns (Shu et al., 2020). The SHUD model's capacity to simulate both surface and subsurface hydrological processes renders it a valuable tool for examining interactions between groundwater and surface water. Additionally, its adaptive time-step (seconds to minutes) feature enhances computa-

tional efficiency by adjusting the time-step in response to the complexity of the simulated processes, thereby ensuring precise representation of both rapid and slow hydrological processes. The details and performance of the SHUD model can be found in Shu et al. (2020).

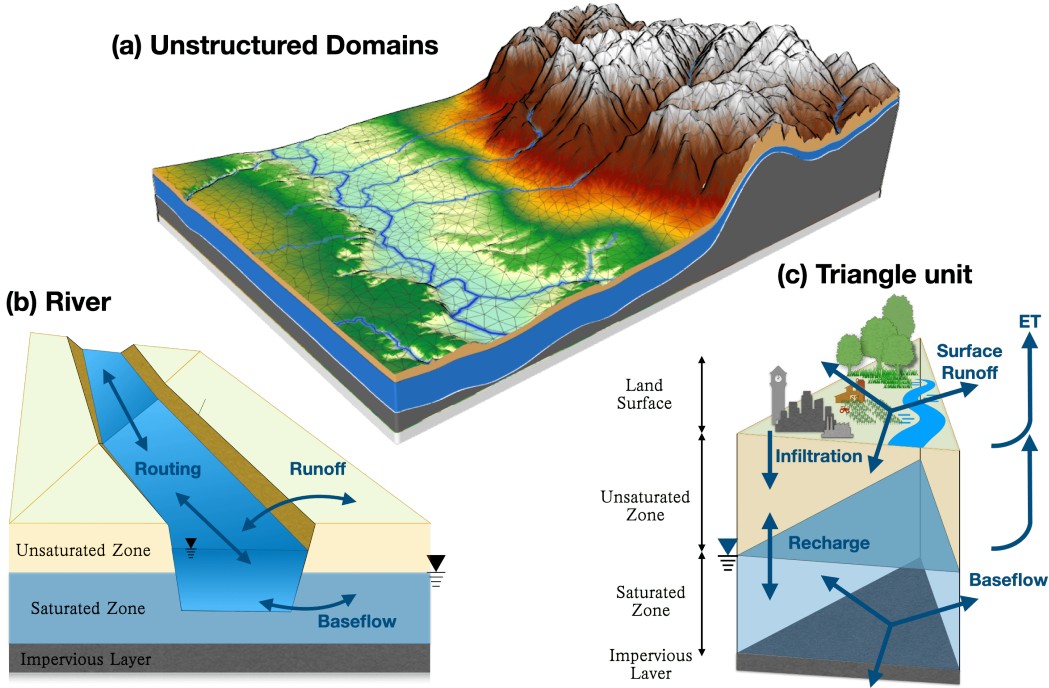

**Figure 2.** The structure of SHUD model; (a) the triangular mesh in watershed scale, (b) the flux exchange between river and hill-slope elements, (c) the three-layer triangular element.

In SHUD V 1.0, there are two types of Hydrological Computing Units (HCUs): triangular slope elements and trapezoid segments for river reaches. Each triangular HCU is further divided into three layers: land surface, unsaturated zone, and saturated zone. Therefore, the total number of HCUs in a watershed model is $N = 3 \times N_{ele} + N_{riv}$, where $N_{ele}$ is the number of triangular elements, and $N_{riv}$ is the number of river reaches. The terminology *element* and *cell* are used interchangeably in our model, both referring to the unstructured triangle.

The primary task of the SHUD model as a numerical hydrological model is to calculate the fluxes among the HCUs. The land surface layer computes snow accumulation/melting, interception, infiltration, and lateral fluxes to triple neighbor elements. The unsaturated zone only calculates vertical infiltration/exfiltration and recharges to the saturated zone. The saturated zone calculates the lateral groundwater flow (or baseflow) among the triangular elements. Both the unsaturated and saturated zone response to the potential evapotranspiration, based on the water content and groundwater level.

To model a lake within a basin context, a model must define the two-dimensional area of the lake domain and the respective boundary conditions, including water levels, inflows, and outflows. Information on the physical characteristics of the lake,

such as its volume, surface area, and depth, must also be provided. Once the lake domain has been established, hydrological processes within and around the lake can be simulated, including precipitation, evaporation, runoff, and groundwater flow. Furthermore, the model simulates the movement of nutrients, sediment, and other substances within the lake. To execute the SHUD model accurately, input data on topography, climate, land use, and soil properties must be provided. This data is crucial for simulating the hydrological processes in the lake and the surrounding landscape. The coupled model output provides a deeper understanding of the lake's hydrology, and different management strategies can be assessed to determine their impact on the lake's ecosystem health and water quality.

## 2.3 Lake Coupling

In the context of the SHUD model, lake coupling involves integrating the lake model with the hydrological model to simulate the lake's water balance and its influence on surrounding hydrological processes. This enhanced model, which couples a lake module with SHUD, treats the lake as a distinct entity within the model domain, complete with its own set of governing equations and boundary conditions. The lake model simulates the hydrological processes occurring within the lake, such as evaporation, inflows, outflows, and water storage. Concurrently, the hydrological model simulates the processes unfolding in the surrounding landscape, including precipitation, interception, evapotranspiration, surface runoff, groundwater flow, and river routing. The model also accounts for the inflows and outflows to and from the lake.

The water balance of the lake is generally defined by the following equation:

$$\frac{\Delta S}{\Delta t} = A_b(P - E) + Q_s + Q_g + R_i - R_o \tag{1}$$

where $\Delta S$ represents the change in storage within the time interval $[L^3]$; $\Delta t$ is the time interval $[T]$; $A_b$ is the area of lake bucket $[L^2]$; $P$ is the precipitation rate $[LT^{-1}]$; $E$ is the actual evaporation on lake surface $[LT^{-1}]$; $Q_s$ is the surface direct runoff from the surrounding land to the lake $[L^3T^{-1}]$; $Q_g$ is the total groundwater flux into the lake $[L^3T^{-1}]$; $R_i$ is the total water fluxes into the lake $[L^3T^{-1}]$; and $R_o$ is the total fluxes from the lake into the downstream rivers $[L^3T^{-1}]$, which is zero for a closed lake. Therefore, the essential task for lake hydrological modeling is to calculate the flux items between the surrounding land and the lake.

### 2.3.1 Hydrological Computing Unit

Within the computational domain of the coupled SHUD model, a lake is conceptualized as a "bucket" (Fig. 3). This "bucket" is geometrically defined by the lake's bathymetric curve, which establishes a relationship between the lake stage and its surface area (Fig. 3 (b)). The bathymetric curve used in the model might be a simplified representation of the actual curve. For each specified lake stage, a corresponding top surface area is determined. Utilizing this curve, we can compute the volume of lake water ($V(y)$) by accounting for the fluxes into and out of the lake over a designated time interval (Eq. 2). Concurrently, the fluctuation in the lake stage ($y$ or $y_{lk}$) can be gauged by simulated changes in the lake's volume ($V(y)$) . However, alterations in the lake's surface area ($A(y)$) do not impact the boundary delineation between the lake and the neighboring land within the

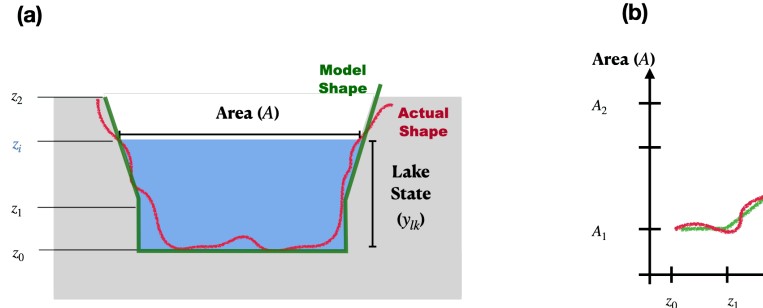

**Figure 3.** Representation of lake geometry in the coupled model. The lake is portrayed as a bucket (a), its form determined by a bathymetry curve (b) that associates lake stage ($y_{lk}$) or elevation ($z$) with surface area.

model. The dynamic expansion and contraction of lakes are not considered within the internal and adjacent elements of the lake domain.

$$V(y) = \int_0^y A(y) \cdot dy \tag{2}$$

In order to enable lake coupling, a novel Hydrological Computing Unit (HCU) that represents the lake has been integrated into the numerical solver. The lake HCU encapsulates the lake itself, with the water fluxes within the HCU representing the cumulative sum of all the triangular elements contained within the lake.

Within the coupled model, the triangular units are classified into land, bank, and lake elements (Fig. 4(a)), the element distinguished by a flag that indicates its type. The computational algorithms diverge significantly among these element types. Land elements conform to the standard triangular elements found in the SHUD model (Shu et al., 2020). Bank elements, which exist in a transitory state between the land and lake, are processed using a specialized method. The surface and subsurface fluxes over the lake-facing edges of a bank element are computed based on the hydraulic disparity between the bank and lake elements.

Conversely, the vertical fluxes, including precipitation and evaporation, are handled within the lake element, while the lateral fluxes (comprising surface and subsurface fluxes between bank and lake elements) are directly addressed within the lake HCU. Notably, although a lake HCU consists of multiple triangular units, the lake's hydrology is considered holistically. This implies that the water stage of each triangular lake element is congruent. Precipitation and evaporation occurring on the lake surface correspond to the summation of the vertical fluxes across the lake units.

### 2.3.2 Fluxes Calculation

The correct synchronization of the lake and hydrological models requires defining the water exchange between the lake and the surrounding landscape, which includes factors like surface runoff and groundwater flow (Fig. 4 (b)). Calculations for these

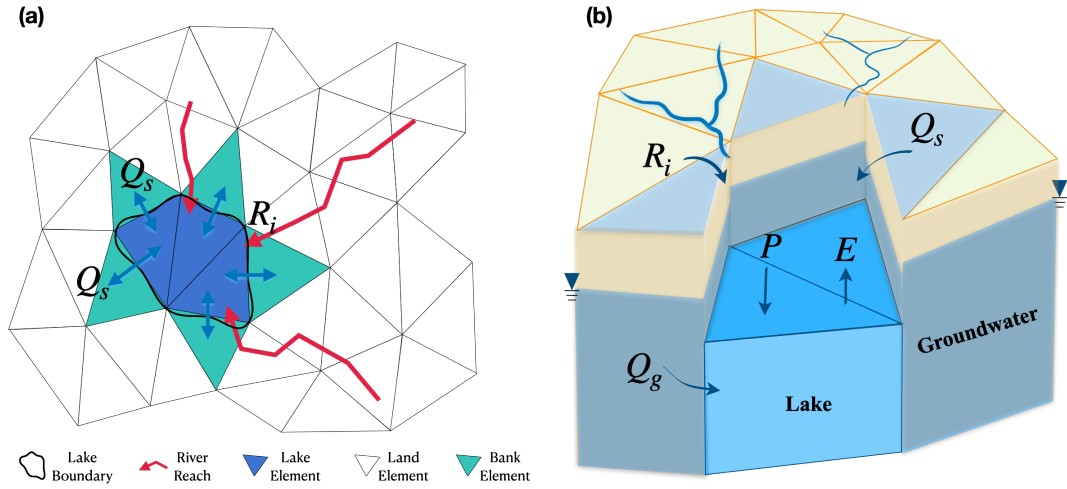

**Figure 4.** Depiction of elements and fluxes within the coupled domains. (a) Illustration of the three types of triangular elements: land, bank, and lake elements. (b) A 3D perspective of these elements along with the fluxes interacting among them.

exchanges start with the lake's water balance (Eq. 1). These fluxes can then be split into the sum of the fluxes on each lake element (Eq. 4).

$$
\frac{\Delta S}{\Delta t} = A_b(P - E) + Q_s + Q_g + R_i - R_o \tag{3}
$$

$$
= \sum_{j=1}^{N_l} P^j A^j + \sum_{j=1}^{N_l} E^j A^j + \sum_{j=1}^{N_b} Q_s^j + \sum_{j=1}^{N_b} Q_g^j + \sum_{j=1}^{N_r} R_i^j + 0 \tag{4}
$$

In this formula, $N_l$, $N_b$, and $N_r$ represent the number of lake elements, bank elements, and river outlets into a lake, respectively. $P^j$ is the precipitation falling into a lake element with an area of $A^j$. Within the SHUD model, the potential evapotranspiration (PET) is calculated using the Penman-Monteith equation, and the actual evapotranspiration (AET) on each lake element is equivalent to the PET on it. The sum of the AET of all lake elements becomes the total AET of the lake.

The total overland runoff to a lake is comprised of the sum of overland fluxes between the bank elements and the lake. In a similar vein, the aggregate groundwater flow from the land to the lake is calculated as the sum of the groundwater fluxes between the bank and lake elements. This calculation is contingent on their respective hydraulic heads, terrains and hydraulic characteristics.

$$Q_s^e = C_{wr} L_e y_{sf}^{\frac{3}{2}} \sqrt{2g} \qquad (5)$$

$$Q_g^e = \overline{K} \cdot \frac{(y_{gw} + z_b) - (y_{lk} + z_b^l)}{d_l} \cdot \left[ \frac{L_e}{2} (y_{gw} + y_{lk}) \right] \qquad (6)$$

$$Q_r = \frac{A_{cs}}{n} \left( \frac{A_{cs}}{p_w} \right)^{\frac{2}{3}} s_r^{\frac{1}{2}} \qquad (7)$$

In these equations, $Q_s^e$ and $Q_g^e$ represent the overland runoff and groundwater flow in $e$ direction ($e = C(1,2,3)$) of a triangular
element, expressed in units of volume per time $[L^3 T^{-1}]$; $Q_r$ stands for the discharge from the river to the lake. The terms $y_{sf}$,
$y_{gw}$, and $y_{lk}$ denote the water head of surface ponding from the land surface, the groundwater head from bedrock, and the lake
hydraulic head or lake stage from the lake-bed, respectively, all expressed in length units $[L]$. $C_{wr}$ is the discharge constant
for weir flow $[-]$, and $n$ represents Manning's roughness $[TL^{-1/3}]$. $L_e$ is the length of a triangle's edge in $e$ direction $[L]$; $\overline{K}$
is the average hydraulic conductivity of a bank element and its adjacent lake element $[LT^{-1}]$; $z_b$ and $z_b^l$ are the elevations of
the bank element bedrock and the lake-bed $[L]$, respectively; $d_l$ is the distance between the centroids of a bank element and its
neighboring lake element $[L]$; $A_{cs}$ is the cross-sectional area of river flow; $p_w$ is the wetting-perimeter in a river channel $[L^2]$;
and $s_r$ is the hydraulic gradient $[LL^{-1}]$.

## 2.4 Numerical solver

In SHUD, the initial value problem for these ordinary differential equations (ODEs) is formulated as follows:

$$\frac{d\boldsymbol{Y}}{dt} = f(t, \boldsymbol{Y})$$

$$\boldsymbol{Y}(t_0) = \boldsymbol{Y_0}.$$

Here, the discrete state vector is denoted by $\boldsymbol{Y}$:

$$\boldsymbol{Y} = \begin{pmatrix} \boldsymbol{Y_{sf}} \\ \boldsymbol{Y_{us}} \\ \boldsymbol{Y_{gw}} \\ \boldsymbol{Y_{riv}} \end{pmatrix}.$$

$\boldsymbol{Y_0}$ represents the initial conditions, and $f(t, \boldsymbol{Y})$ denotes the equations governing the hydrologic flow.

In the coupled model, the $\boldsymbol{Y}$ value is updated with the added $\boldsymbol{Y_{lk}}$, so the new $\boldsymbol{Y}$ is:

$$\boldsymbol{Y} = \begin{pmatrix} \boldsymbol{Y_{sf}} \\ \boldsymbol{Y_{us}} \\ \boldsymbol{Y_{gw}} \\ \boldsymbol{Y_{riv}} \\ \boldsymbol{Y_{lk}} \end{pmatrix}.$$

The $\boldsymbol{Y}_{lk}$ represents the lake stage of all lakes in the model domains. For instance, if there are three lakes in the domains, then $\boldsymbol{Y_{lk}}$ is:

$$\boldsymbol{Y_{lk}} = \begin{bmatrix} y_{lk}^1 \\ y_{lk}^2 \\ y_{lk}^3 \end{bmatrix}.$$

The change in the lake stage, $d\boldsymbol{Y_{lk}}$, is:

$$d\boldsymbol{Y_{lk}} = \begin{bmatrix} dy_{lk}^1 \\ dy_{lk}^2 \\ dy_{lk}^3 \end{bmatrix}.$$

In this equation, $dy_{lk} = \frac{\Delta S}{A(y)}$, where $A(y)$ is the lake top area when lake stage equates $y$, which is based on the defined lake bathymetry curve. The length of both vectors $\boldsymbol{Y}$ and $d\boldsymbol{Y}$ equals $N$, where $N = 3 \times N_{\text{ele}} + N_{\text{riv}} + N_{\text{lk}}$. The $N$ is the total length
of HCUs in the coupled model.

In the coupled model, lake elements serve as agents for flux calculation, with fluxes computed based on the properties of these elements. However, the mass balance of the lake element is omitted, and the fluxes are instead incorporated into the lake's water balance. As a result, the $y_{sf}$, $y_{us}$, and $y_{gw}$ values of the lake elements remain constant in the vectors $\boldsymbol{Y_{sf}}$, $\boldsymbol{Y_{us}}$, and $\boldsymbol{Y_{gw}}$. These unvarying values, albeit constant, do not impede the calculations or the efficiency of the numerical solver and remain at
zero in the $d\boldsymbol{Y}$ vector.

Upon completion of the new vectors $\boldsymbol{Y}$ and $d\boldsymbol{Y}$, the task of handling iterations and outputting the results of each step is transferred to the CVODE numerical solver (Hansen, 2016), a tool developed by the Lawrence Livermore National Laboratory (LLNL) for initial condition problems. The initial condition of the lake, indicating the lake stage at the outset of the simulation, is stored in conjunction with the initial conditions of other HCUs in the *.cf.ic* files.

## 2.5  Data

The SHUD modelling framework employs three distinct categories of data: meteorological forcing data, terrestrial data, and observational data. The latter serves to facilitate model calibration and the configuration of initial conditions.

Meteorological forcing data were sourced from the China Meteorological Forcing Dataset (CMFD) (He et al., 2020) due to the limited availability of in-situ meteorological stations within the study area. The CMFD incorporates variables such
as precipitation, air temperature, specific humidity, short-wave radiation, wind speed, and air pressure, providing sufficient variables to drive the SHUD model. Despite the CMFD's extensive coverage of China and high-quality reanalysis, it has an inclination to underestimate rainfall intensities, which can potentially lead to an underestimation of peak stream discharge. The dataset is characterised by a 6-hour time interval and a 0.1-degree horizontal resolution, resulting in a total of 386 CMFD grids within the simulation domains (Fig. 5).
Terrestrial data was gathered from the Global Hydrological Data Cloud (https://shuddata.com, accessed June 10, 2023). The Digital Elevation Model (DEM) was derived from the ASTER Global DEM (NASA et al., 2018), providing a resolution of 3-arc-seconds (approximately 30 meters). Soil classification and texture data were extracted from the Harmonized World Soil

Database v1.2 (Nachtergaele et al., 2008), offering a 1-km resolution. The 0.5-km USGS MODIS land cover dataset (Broxton et al., 2014) was the source of land cover data, integral to the model deployment process.

Observational streamflow data was acquired from the Buha gauge station (see the location in Figure 1). This dataset, encompassing daily streamflow measurements from 1980 to 2017, was obtained from a local gauge station and is currently not publicly accessible. Hydrological data specific to Qinghai Lake was sourced from the Qinghai Lake hydrology-meteorological dataset (Zhang, 2021). The lake, with an average area of 4360 $km^2$ and an average lake level of 3195 m, recorded its lowest level of 3192.86 m in 2004. This dataset (Zhang, 2021) provides invaluable insights into the hydrological dynamics of Qinghai Lake, contributing to a nuanced understanding of the lake's hydrological and environmental significance.

## 2.6 Model Deployment

The SHUD model was deployed using the rSHUD package and the AutoSHUD script, with the corresponding R script and terrestrial data accessible as per Shu et al. (2023); Shu (2023b).

The domain decomposition results for the Qinghai Lake Basin (QLB) consists of 4773 triangular elements, of which 688 are designated as lake elements and 785 as bank elements (Figure 5). The domains encompass a total lake area of 4404 $km^2$, with the surrounding watershed area accounting for 25210 $km^2$. While Figure 5 might seem similar to Figure 1 at first glance, it specifically represents the unstructured triangular mesh model domains constructed by the rSHUD tool (Shu et al., 2023), highlighting the computational framework applied in our study. The average area of the triangular elements within the model domain is 6.2 $km^2$. The river network within the domain, spanning a total length of 4122 km, features an average river reach length of approximately 2.5 km. The model incorporates 1633 river reaches and 45 river outlets, all feeding into the lake.

## 2.7 Calibration

The model's initial condition is established using the state after a preliminary 40-year simulation (from 1979 to 2018). Specifically, the state at the end of this simulation period (December 21, 2018) is employed as the initial condition for both calibration and long-term simulation. The initial lake stage is set 25 m above the lowest point of the lake bed, corresponding to a lake level of 3183 m.

Model calibration was performed using the Covariance Matrix Adaptation - Evolution Strategy (CMA-ES) (Hansen, 2016). This evolutionary algorithm produces 96 offspring in each generation, maintaining the best-performing offspring as the seed for the subsequent generation with additional perturbations. These perturbations are derived from the covariance matrix of the previous generation. The calibration period extends from 2004 to 2008, with two validation periods: 1980-1999 and 2009-2017. The first validation period is 20 years prior to the calibration period, while the second follows 9 years after. The calibration begins with a CMA-ES calibration for the period 2002-2008, using 2002-2003 as the spin-up period. After 14 generations of CMA-ES iterations, the calibration tool identifies an optimal parameter set. This parameter set is then utilized for the 40-year simulation, after which the model's performance is analyzed.

The model's ability to simulate daily and monthly discharges in the Buha River is demonstrated in Figures 6 and 7 respectively. During the calibration period, three goodness-of-fit (GOF) indicators - Nash-Sutcliffe efficiency (NSE, Nash and

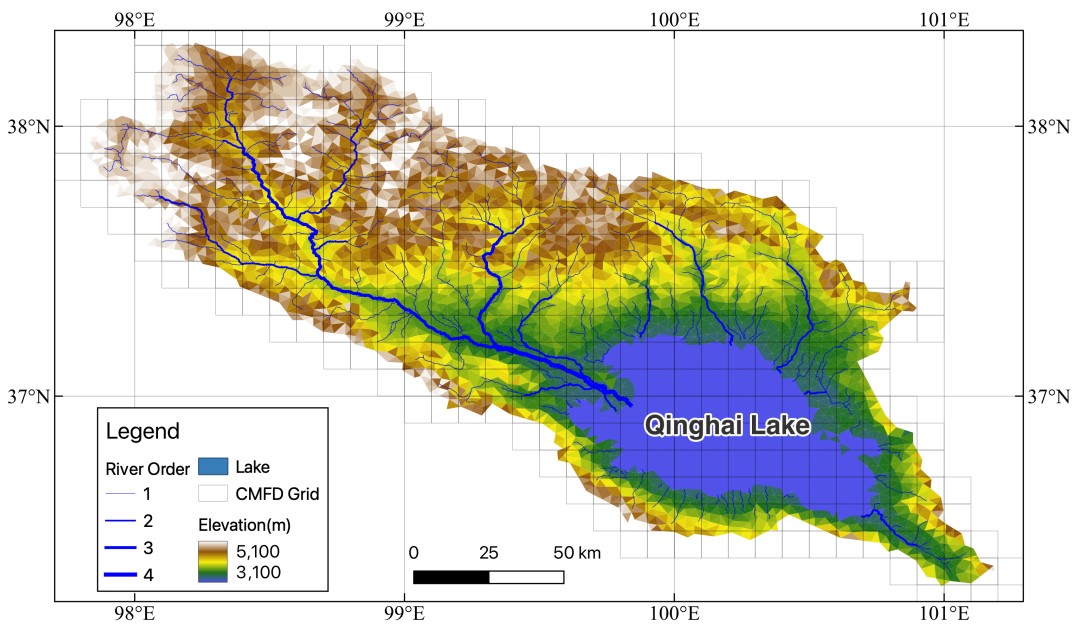

**Figure 5.** The rSHUD-constructed unstructured domains in the Qinghai Lake Basin, comprising 4773 triangular elements of which 688 are designated as lake elements and 785 as bank elements.

Sutcliffe (1970)), Kling-Gupta Efficiency (KGE, Gupta et al. (2009)), and Determination Coefficient ($R^2$) - yield values of 0.62, 0.58, and 0.65, respectively (Fig. 6). The GOF of daily streamflow in the validation period is lower than that in the calibration period (as expected), with NSE = 0.41/0.46, KGE = 0.44/0.52, and $R^2$ = 0.46/0.47. As anticipated, the monthly GOFs are higher than the daily GOFs, with NSE = 0.76, KGE = 0.64, and $R^2$ = 0.82 for the calibration period, and NSE = 0.56/0.56, KGE = 0.54/0.59, and $R^2$ = 0.63/0.57 for the validation periods.

## 3 Results

In this study, we primarily aim to simulate lake dynamics by employing the developed lake-watershed coupled model. We have previously presented the simulation results of stream flow in the calibration section, and hence, this section centers on lake-specific outcomes, namely the alterations in lake level and the components of the lake's water balance.

### 3.1 Lake level

Figure 8 delineates the simulated temporal variations in lake level. Our observational dataset extends from 1960 to 2020, however, the simulation period is confined to 1979-2018, conditioned by the availability of CMFD data. We define the mean lake level in 2000 as the benchmark level, and all changes in lake level are computed relative to this datum point. In multi-

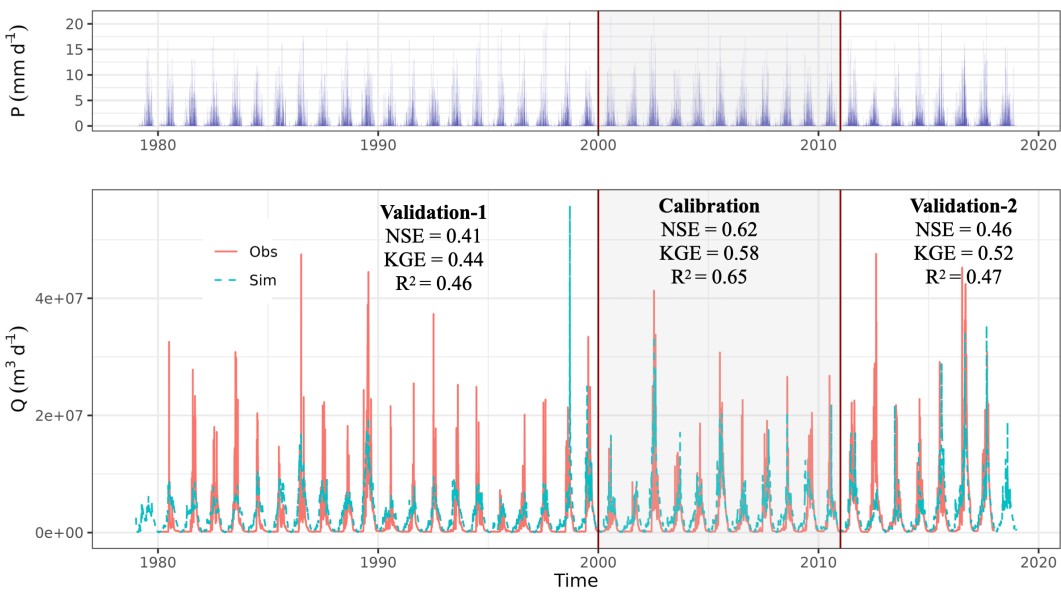

**Figure 6.** The hydrograph of daily discharge in Buha River during calibration and validation periods.

decades simulations, the change in lake water mass is more valuable and reliable than elevation above sea level. According to
the lake settings, the lake is depicted as a bucket determined by lake stage and top area. Without detailed bathymetry (function
of lake stage and top area), we can only roughly describe the lake shape (Figure 3). The rough estimation of lake bathymetry
introduces an error in simulating lake surface elevation. Moreover, the DEM and lake level measurements, sometimes based on
different datum, exhibit discrepancies. The key target in lake hydrology is not the water level elevation, but the change in lake
water volume, i.e., the lake water balance. In the results, the changes in lake water volume are more reliable than the absolute
values of lake water level in the coupled model, hence Figure 8 compares the fluctuations in water level.

In general, the simulated lake level variations exhibit a high degree of alignment with the observed changes, tracing a
decrease-increase U-shaped trajectory. Nonetheless, a minor underestimation of the lake level is observed at the begining of
the simulation period (Fig. 8 (a)). The line-fit plot (Fig. 8 (b)) offers further insight into the model's proficiency in replicating
lake level alterations, with GOF values of NSE = 0.71, KGE = 0.63, and $R^2$ = 0.77.

**3.2  Water balance of the lake**

The perspective of water balance is indispensable for deciphering the hydrological attributes of a lake within a basin. As
outlined in Eq. 1, the water balance is partitioned into six constituents: fluctuations in lake storage $(\Delta S)$ , precipitation $(P)$
, evaporation $(E)$ , and contributions from rivers $(R_i)$ , surface runoff $(Q_s)$ , and groundwater flow $(Q_g)$ . Throughout the

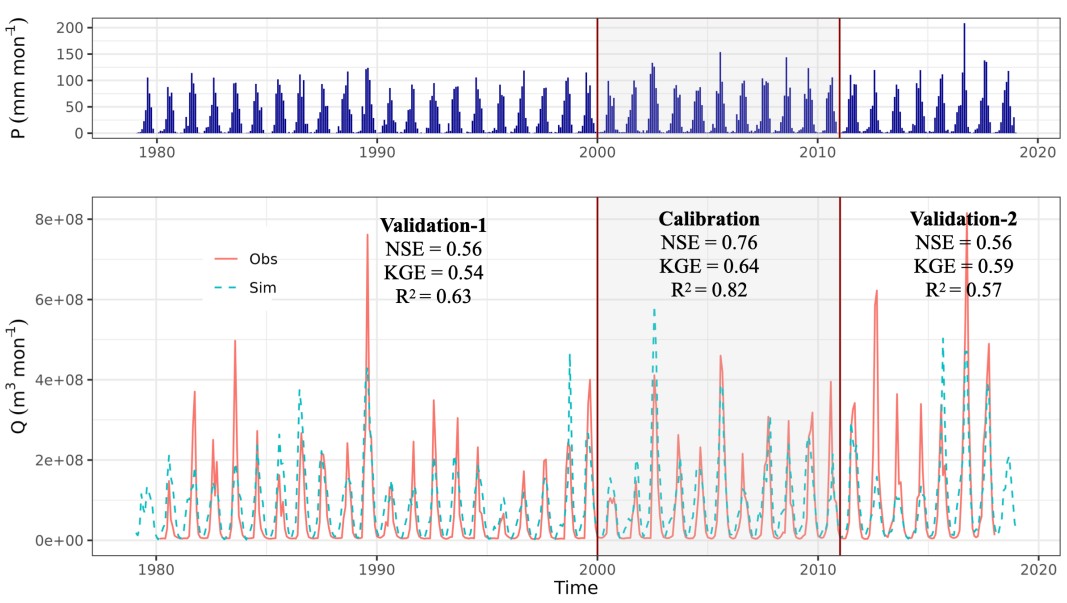

**Figure 7.** The hydrograph of monthly discharge in Buha River during calibration and validation periods.

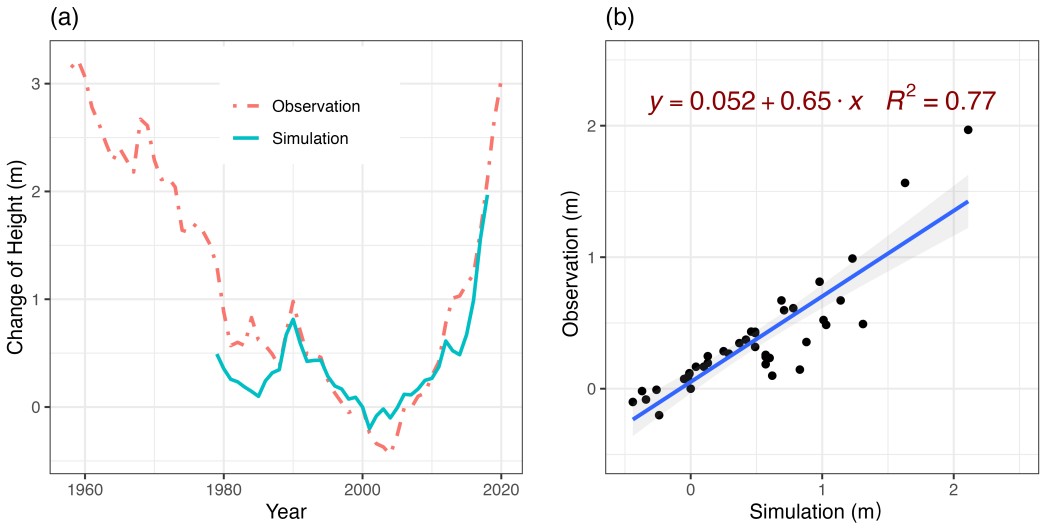

**Figure 8.** Changes in lake level, observation vs simulation, relative to the mean lake level in 2000 as the reference level. (a) Comparison of lake level changes; (b) Line-fit plot of simulated and observed values.

simulation period (1979-2018), the lake level escalated by 1696 mm, a balanced outcome derived from the interplay of the other five components. The mean annual precipitation ($P$) and evaporation ($E$) are approximately 386 mm and 996 mm, respectively, with the latter aligning with values reported in the literature (Dong et al., 2019; Su et al., 2019). The annual precipitation over the lake from the CMFD dataset is less than the basin average precipitation. Contributions from rivers ($R_i$) amount to about 587 mm annually, while surface runoff ($Q_s$) and groundwater flow ($Q_g$) contribute a relatively minor 17 mm and 28 mm per year, respectively (Table 1).

**Table 1.** Annual mean water balance components in Qinghai Lake from 1979 to 2018 as simulated by the coupled model.

| Component | $\Delta S$ | $P$ | $E$ | $R_i$ | $Q_s$ | $Q_g$ |
|---|---|---|---|---|---|---|
| Annual mean ($\times 10^8 \mathrm{m}^3$) | 1.70 | 15.44 | 39.83 | 23.47 | 0.67 | 1.13 |
| Annual mean (mm) | 42 | 386 | 996 | 587 | 17 | 28 |
| Percentage to $P$ (%) | 11.0 | 100 | 257.9 | 152.0 | 4.3 | 7.3 |

During the simulation period, with precipitation ($P$) as 100 units, evaporation ($E$) accounts for 258, river flux ($R_i$) in at 152, and surface ($Q_s$) and groundwater($Q_g$) fluxes are 4 and 7 respectively. This sums up to a lake water mass increase ($\Delta S$) of 11 units approximately. The contributions from surface and groundwater total 11 units, indicating that excluding these contributions would result in non-increase in lake water. Though relatively small, the contributions from surface and groundwater are not negligible. Additionally, the mentioned surface and groundwater fluxes from SHUD model are only those entering the lake directly through its boundaries. Due to topographical factors, land surface and subsurface fluxes typically first enter rivers, then flow into the lake through these rivers. Areas directly contributing surface and subsurface fluxes to the lake without passing through rivers are very limited, usually confined to small sections along the lakeshore. Due to differences in the algorithms used for water balance partitioning, our simulation results vary from those reported in other literature, with the literature indicating significantly higher groundwater contribution compared to our simulations (Li et al., 2007; Zhang et al., 2011; Cui and Li, 2016).

Six major rivers flow into Qinghai Lake: the Buha, Shaliu, Quanji, Haergai, Daotang, and Heima rivers. Collectively, they contribute approximately $15.94 \times 10^8 m^3/a$, which represents 66.8% of the lake's total runoff of $23.48 \times 10^8 m^3/a$ (Table 2). The catchment areas of these six rivers encompass 78% of the land area, contributing 67.9% of the total runoff to Qinghai Lake. Notably, the Buha River is the largest contributor, accounting for 46.7% of the total runoff into the lake.

Upon comparing our simulated results with historical literature (Li et al., 2007; Zhang et al., 2011; Cui and Li, 2015b, 2016; Zhang, 2021), we observe that while our findings are reliable, they exhibit discrepancies with previously reported data. The literature indicates a total river runoff recharge of approximately $17.78 \times 10^8 m^3/a$, with five rivers accounting for an annual runoff of $14.28 \times 10^8 m^3/a$. Although the contributions from the major rivers are comparable, the total runoff recharge reported in the literature significantly differs from our findings, leading to notable variances in the contribution percentages. The total runoff data in these publications are based on a 1994 report (Lanzhou Branch of Chinese Academy of Sciences, 1994). In contrast, recent provincial water resources bulletins (Qinghai Provincial Department of Water Resources, 2022) suggest a

**Table 2.** Long-term average contributions of major rivers to Qinghai Lake and their respective percentages of total inflow, with comparable values from other literature.

| Name | Area (km$^2$) | Modeling Q ($\times 10^8$m$^3$) | % | Li et al. (2007)(1959-2000) Min | Max | Average | % | Cui and Li (2015a, 2016)(1960-2010) Average | % |
|---|---|---|---|---|---|---|---|---|---|
| Buha R. | 14932 | 10.96 | 46.7% | 1.99 | 16.63 | 7.85 | 48.7% | 8.09 | 45.5% |
| Quanji R. | 599 | 0.72 | 3.1% | 0.079 | 0.86 | 0.55 | 3.4% | 0.54 | 3.0% |
| Shaliu R. | 1645 | 1.68 | 7.1% | 1.05 | 3.92 | 2.46 | 15.3% | 3.12 | 17.5% |
| Haergai R. | 1572 | 1.99 | 8.5% | 1.96 | 3.35 | 2.42 | 15.0% | 2.42 | 13.6% |
| Daotang R. | 818 | 0.33 | 1.4% | - | - | - | - | - | - |
| Heima R. | 123 | 0.26 | 1.1% | 0.016 | 0.41 | 0.11 | 0.7% | 0.11 | 0.6% |
| Major river | 19689 | 15.94 | 67.9% | 5.095 | 25.17 | 13.39 | 83.1% | 14.28 | 80.3% |
| Land Area | 25210 | 23.48 | 100% | - | - | 16.12 | 100% | 17.78 | 100% |

multi-year average runoff in the Qinghai Lake basin ranging between 22.2-26.7$\times 10^8 m^3/a$, aligning more closely with our simulation results. These variances underscore the necessity for further research.

Despite discrepancies, our simulation underscores the need for further research and confirms the model's relevance for lake basin studies. While it's challenging to claim new findings for Qinghai Lake research from our limited data analysis, our model emerges as a valuable tool for future research, offering a robust framework to enhance understanding of the lake's hydrology.

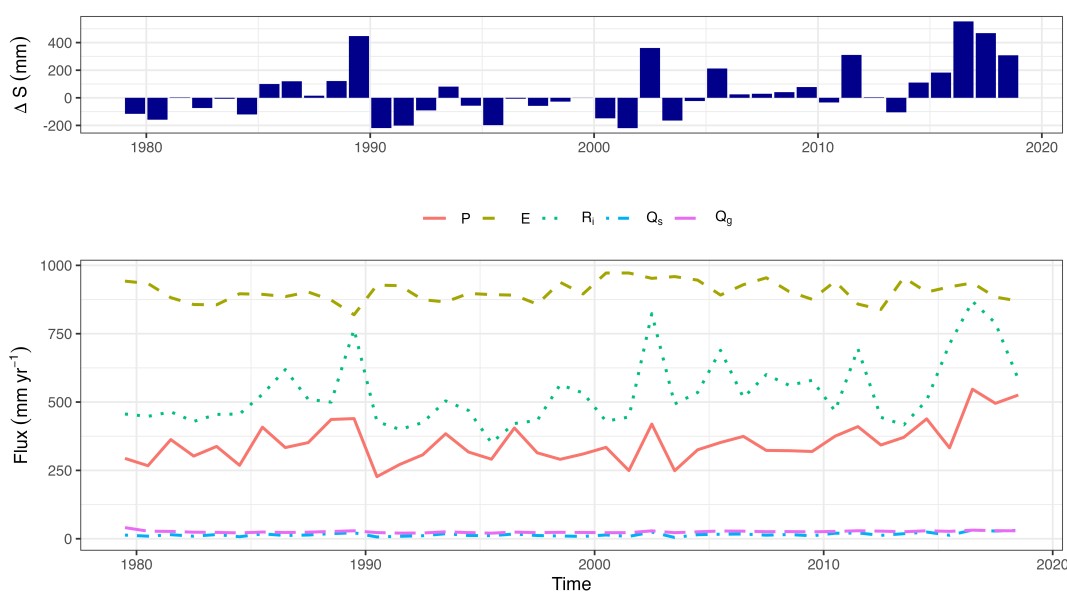

**Figure 9.** The water balance of Qinghai Lake, including six components: change in lake storage ($\Delta S$), precipitation ($P$), evaporation ($E$), and contributions from rivers ($R_i$), surface runoff ($Q_s$), and groundwater flow ($Q_g$).

## 4 Discussion

It is noteworthy that the current lake coupling scheme is specifically designed for closed lakes, as the algorithm to handle downstream outlets has yet to be incorporated, though this limitation is not insurmountable and will be addressed in future updates. The model presently implies that the calculated top surface area of the lake does not influence the adjacent land elements. In essence, land elements remain static and do not transition into lake elements when the lake expands; the number of lake elements is maintained as constant. Higher levels of lake dynamics complexity are typically accommodated by employing 2D or 3D computational fluid dynamics (CFD) or hydrodynamics which offer a more granular understanding of lake behaviors and superior spatial resolution compared to hydrological models (Munar et al., 2018). As such, we underscore that the coupled model contributes invaluable data on rivers, surface, and subsurface flows at the basin scale, furnishing detailed boundary conditions integral for further studies in hydrodynamics, limnology, or biogeochemical cycles (Cobourn et al., 2018; Ladwig et al., 2021).

We acknowledge that the GOF measures for streamflow in the calibration and validation periods could be enhanced with additional model improvements. Upon scrutiny of the time-series of streamflow, we observed that the present hydrological model falls short in simulating the perennial and seasonal permafrost prevalent in this region. The high-altitude area of the QLB is characterized by perennial permafrost, while the regions surrounding the lake feature seasonal permafrost. The dynamism of permafrost exerts a substantial influence on hydrological processes, particularly on surface and subsurface flows. In the cold season, we noted that river discharge tends to approach zero, with distinct freeze-thaw phases, but the simulated streamflow fails to accurately capture these dynamics. This shortcoming is attributable to the hydrological modeling rather than the lake-watershed coupling study. Nevertheless, it highlights the need to devise a novel algorithm for permafrost dynamics when deploying the model in cold regions. Additionally, the utilization of data more reliable than reanalysis data, which often underestimates heavy rainfall and rainfall intensity, could potentially improve the GOF measures.

We observe that the simulated lake level underestimates the measurements at the onset of the simulation period (1979-1987), a discrepancy likely stemming from low initial conditions for the lake level. Given the pronounced impact of the initial lake level on the ultimate outcome, it is imperative to exercise rigorous methodological consistency when setting the initial conditions of the lake in the modeling process.

## 5 Conclusion

In this study, we have introduced a novel methodology for coupling lake-watershed models and showcased its implementation in the Qinghai Lake Basin. Our model capably simulates lake level dynamics and offers an all-encompassing analysis of the lake's water balance. The model's ability to faithfully replicate observed changes in lake level and streamflow at the Buha River station underscores its potential value for hydrological studies in regions abundant with lakes.

The model provides fresh insights into lake hydrology, distinguishing the contributions of various components such as river inflow, surface runoff, and groundwater flow. This level of detail, unachievable solely through observational data, augments our understanding of hydrological processes occurring in lake basins. However, it should be noted that the current scheme is limited

to closed lakes, and it does not take into account the dynamic expansion and contraction of lakes. These limitations are set to be addressed in future updates, further augmenting the model's versatility. The model's performance could be elevated further by incorporating more reliable data sources and refining the model's algorithms. A particular area for future enhancement is the simulation of permafrost dynamics, which significantly influence hydrological processes in cold regions.

As a hydrological model, SHUD primarily focuses on watershed hydrology and the horizontal water exchanges between
355 the lake and its surrounding land. It employs a simplified potential evapotranspiration scheme to address the energy fluxes and atmosphere-land interactions often examined in land surface model (such as CLM and NOAH). Therefore, the proficiency of land surface models in vertical energy coupled with SHUD's expertise in horizontal hydrological processes hints at a potential for coupling. Such coupling could furnish a more refined depiction of water and energy storage and movement within lakes and watersheds.

In conclusion, the lake-watershed fully-coupled model developed in this study signifies a substantial advancement in the realm of hydrological modeling. While there are areas that necessitate further improvement and expansion, the model presents a potent tool for understanding and predicting hydrological processes in lake basins. Its successful deployment in the Qinghai Lake Basin illustrates its potential for wider application in hydrological studies and water resources management.

*Code and data availability.* All the source code and data in this paper, are saved in Zenodo (Shu, 2023a), except the observational streamflow
in Buha River Gauge Station.

In addition to the Zenodo repository, we also store all source code on Github.

SHUD model: https://github.com/SHUD-System/SHUD (Shu et al., 2020).

rSHUD package: https://github.com/SHUD-System/rSHUD (Shu et al., 2023).

AutoSHUD: https://github.com/SHUD-System/AutoSHUD.

*Author contributions.* L. Shu and X. Li – Conceptualization, Investigation, Methodology, Software, Validation, Visualization, Writing original draft and editing. Y. Chang, X. Meng, H. Chen, S. Lyu – Supervision, Investigation, Writing original draft and editing. Z. Li, H. Wang, Y. Qi.– Data preparation, Code development, program test, editing.

*Competing interests.* The authors declare that no competing interests are present.

*Acknowledgements.* This study is funded by the National Natural Science Foundation of China (41930759), West Light Foundation of the
375 Chinese Academy of Sci ences (xbzg-zdsys-202215), National Natural Science Foundation of China (41961012), the Chinese Academy Sciences Talents Program, the Qinghai Key Laboratory of Disaster Prevention (QFZ-2021-Z02), National Cryosphere Desert Data Center

(E01Z790215), Gansu Provincial Science and Technology Program (22ZD6FA005), and Lanzhou Science and Technology Plan Projects in 2023 (2023-1-49).

The GPT-4 (https://chat.openai.com) only assisted in language refinement.

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
