# Peer review of "Advancing Understanding of Lake-Watershed Hydrology: A Fully Coupled Numerical Model Illustrated by Qinghai Lake"

_Hydrology and Earth System Sciences, 2023_

## Author Response (AR1)

**Reply to Comment #1.**

Following are my reply to the comments.

The reviewer's comments and questions are in bold, and my reply is in blue and normal text.

**This work presented the development and evaluation results of a lake-watershed coupled model, called SHUD. The SHUD model utilizes unstructured triangles as fundamental Hydrological Computing Units, and the coupling between lake and watershed units is considered by calculating the groundwater and surface water recharges. In general, the model is useful to the current literature and provides a powerful tool for understanding and predicting hydrological processes in lake basins. However, there are still some issues needed to be clarified to make the paper clearer and more innovative.**

1. **The innovation of the SHUD model should be clarified. Is there any innovation in the model development? Or the lake model and the interactions between lake and surrounding grids are similar with other land models (e.g., CLM5)?**

Thank you for your comment.

The SHUD model is an integrated surface-subsurface numerical hydrological model (ISSNHM). The merits of ISSNHMs lie in their temporal-spatial continuum, contrasting with other models like SWAT, TOPMODEL, HBV etc. SHUD employs the finite volume method to solve hydrological partial differential equations on unstructured domains. The detailed innovation of the SHUD model and its performance on watersheds are discussed in the model description paper by Shu et al. (2020).

In the lake-coupling scheme, the lake is also decomposed into triangular mesh domains, and the surface, subsurface, and channel fluxes between the lake and surrounding land are calculated.

The lake schemes in CLM and SHUD model are markedly different (based on the CLM 5.0 technology note):

1.  In CLM, the lake is a fraction in a grid cell, described by its depth, extinction coefficient, and fetch, devoid of a physical geometry. Conversely, SHUD-Lake represents the lake as a polygon within a watershed, comprising multiple triangular cells. The lake's volume is a function of its stage and top area.

2.  CLM primarily focuses on vertical energy fluxes, particularly the temperature distribution along depth, plus snow accumulation and melt. On the other hand, SHUD only considers the energy term of evapotranspiration.

3.  Hydrological aspects are scarcely considered in CLM, portraying the lake hydrology as an impervious non-vegetated unit with a constant water mass, only considering snow hydrology in the lake module. In contrast, SHUD represents comprehensive fluxes between lake cells and land cells, via surface, subsurface, and river reaches.

As a land-surface model, CLM emphasizes vertical energy exchanges between the atmosphere and land surface. As a hydrological model, SHUD-Lake concerns the horizontal water exchanges between the lake and its surrounding land.

The strength of CLM in vertical energy and the strength of SHUD in horizontal hydrological processes suggest a potential for coupling, which could provide a better description of water and energy storage and movement in lakes and watersheds.

We have revised the manuscript to elaborate on the SHUD model and the potential coupling with land surface models.

4. **Does the SHUD model consider the subsurface lateral flow (e.g., groundwater flux) between all the land grids? Or does the current model only consider the lateral subsurface water exchange between lake and the surrounding bank grids? If yes, then the coupling between lake and watershed may be limited from the perspective of groundwater exchange. For example, when the model is applied to 1km or finer spatial resolution, the grids that close to the lake (e.g., 2km) may also have influences on the lake.**

Thank you for your comment.

Indeed, the SHUD model thoroughly considers lateral groundwater fluxes between all land triangular grids. The fluxes are calculated based on hydraulic gradients and mean hydraulic conductivities among a cell and its neighboring cells, as detailed in Shu et al. (2020). Thus, this manuscript primarily focuses on the fluxes among lake, bank, and land cells.

You aptly noted that groundwater fluxes between the lake and surrounding land are bi-directional, depending on the hydraulic gradient between the lake and land. The groundwater head around a lake generally aligns with the lake level, slowing groundwater flow and potentially creating wetlands in flat surrounding lands due to groundwater tables being close to the land surface. In more complex scenarios, land groundwater may recharge into the upstream lake edge, while the lake discharges into land groundwater on the downstream lake edges. SHUD-Lake adeptly captures the groundwater head distribution and flux field around the lake as well as across the entire basin, under sufficient spatial resolution.

5. **The figure 9 seems to show that, subsurface groundwater exchange between lake and band grids and the surface runoff are much smaller than others. I wonder whether we can get similar simulation result when we only consider some simple processes as most lake models do (e.g., precipitation, ET, inflow and outflow). If so, some discussions are needed to illustrate the necessity to consider these small terms.**

Thank you for your comment.

Yes, we have analyzed the water balance. Over the past 40 years, with precipitation as 100 Units, evaporation accounts for 253, River flux in at 153, and surface and groundwater fluxes are 4 and 7 respectively. This sums up to a lake water increase of 16 units. The contributions from surface and groundwater total 11 units, indicating that excluding these contributions would result in a smaller increase in lake water. Though relatively small, the contributions from surface and groundwater are not negligible. Additionally, the mentioned surface and groundwater fluxes are only those entering the lake directly through its boundaries. Due to topographical factors, land surface and subsurface fluxes typically first enter rivers, then flow into the lake through these rivers. Areas directly contributing surface and subsurface fluxes to the lake without passing through rivers are very limited, usually confined to small sections along the lakeshore.

We have incorporated this discussion in the revised manuscript.

6. **Why does the Figure 8 compare the anomaly time series? What about the absolute water level?**

Thank you for your comment.

In short-term simulations, the change in lake water mass is more valuable and reliable than elevation above sea level. According to the SHUD-Lake model settings, the lake is depicted as a bucket determined

by lake stage and top area. Without detailed bathymetry (function of lake stage and top area), we can only roughly describe the lake shape (Figure 3). In this paper, the bathymetry for Qinghai Lake is described as the following table:

| Elevation (m) | Top Area (km2) |
| --- | --- |
| 3150 | 4186 |
| 3160 | 4186 |
| 3230 | 4543 |

The initial lake stage value in the simulation is 25 meters, implying an initial lake level of 3175 meters in elevation. The rough estimation of lake bathymetry introduces an error in simulating lake surface elevation. Moreover, the DEM and lake level measurements, sometimes based on different datum, exhibit discrepancies.

However, the key target in lake hydrology is not the water level elevation, but the change in lake water volume, i.e., the lake water balance. In the SHUD-Lake model results, the changes in lake water volume are more reliable than the absolute values of lake water level, hence Figure 8 compares the fluctuations in water level.

We have included this discussion in the section 3.1 of the manuscript.

**7. The figure caption of Figure 9. The "Qb"seems to be "Qg"**

Thank you for your comment.

The correct notation should indeed be "Qg" in the caption. This typographical error has been rectified in the revised manuscript.

**Reply to Comment #2.**

Below are my responses to the reviewer's comments.

The reviewer's comments and questions are highlighted in bold, while my replies are presented in blue, in standard text format.

**In this study, the authors developed a novel lake-watershed coupled model, an enhancement of the Simulator of Hydrological Unstructured Domains (SHUD) for hydrological modelling. Qinghai Lake, a largest salt lake, also an endorheic lake, in the Tibetan Plateau was selected to validate/test the performance of the model. The results show that the model successfully simulates the discharge of the Buha River, a tributary of Qinghai Lake, and quantifies the contribution of the lake water balance. Overall, this manuscript is novel, and was well written. I recommend the publication of this manuscript for publication in HESS after some improvements.**

**Major comments:**
1) **In the Results section, the authors provide some limited information in the "Water balance of the lake". The further quantitative information could be provided, such as the increase/decrease of lake volume in total/sub-period, and the component contribution such as precipitation, evaporation, groundwater and other changes by %.**

Thank you for your suggestion. We revised the statement of water balance of the Qinghai Lake in section 3.2, as **Line 281 - 342.**

**2)** **The quantitative information of lake water balance in Qinghai Lake has been reported by many previous studies. A table can be provided to compare the previous studies with this study. The advantage of this model could be emphasized. What new understanding that this model can provide compared to previous studies or models.**

Thank you for your valuable suggestion.

In response to your comment, we have added a comparison of our simulation results with those reported in previous studies in our revised manuscript. Overall, the differences in the simulation results are not substantial. The most notable discrepancies are in (1) the total inflow to the lake and (2) the contribution of groundwater.

Previous studies by Cui and Li (2015, 2015a, 2016) and Li (2007) adopted data published in 1994, which estimated the total runoff into the lake at 17.78 x10^8 m³/a. However, according to the 2021 Qinghai Province Water Resources Bulletin (Qinghai Provincial Department of Water Resources, 2022), the long-term average inflow into the lake is estimated to be between 22.2-26.7 x10^8 m³/a. Given its more recent survey, this latter data is considered more reliable, and our simulation results are closer to these estimates.

Our coupled model calculates the groundwater inflow into the lake only through the lake's adjacent elements, which are relatively few in number. Groundwater under distant elements enters through the river system before reaching the lake. Therefore, our calculated groundwater contribution to the lake is lower than the estimates in other literature.

While it's challenging to claim new findings for Qinghai Lake research from our limited data analysis, our study demonstrates that our model is a reliable tool for future research on the lake.

We have updated our manuscript to include these discussions on water balance issues in the revised version, as **Line 281 - 342.**

**Specific comments:**

**- "Qinghai Lake" could be added to the title.**

The suggestion is accepted. Since the Qinghai Lake is an exemplary application of the new model, the revised title is "Advancing Understanding of Lake-Watershed Hydrology: A Fully Coupled Numerical Model Illustrated by Qinghai Lake"

**- Introduction. An introduction explaining why Qinghai Lake was chosen could be added.**

Thank you for your suggestion. In response, we have revised the introduction to concisely explain our choice of Qinghai Lake as the study site:

"In this study, we aim to develop and validate a fully coupled lake-watershed hydrological model, with Qinghai Lake in China serving as the primary test site. Qinghai Lake, being the largest lake in China, offers unique hydrological and environmental characteristics that make it an ideal location for our research. Its endorheic nature simplifies the interactions between the lake and its watershed, which is beneficial for testing our model. The wealth of existing research data on Qinghai Lake's hydrology, ecology, and climate is invaluable for both calibrating and validating our model. Furthermore, the lake's high-altitude, cold, and arid conditions make it a representative case study for similar ecosystems, thereby enhancing the model's potential for interdisciplinary research and extending its relevance to other similar environments."

**- The units after the number should be upright letter rather than italic.**

Thank you for your feedback. We have corrected the unit formatting to upright letters throughout the manuscript in the revised version.

- **L75: Suggested reference:
  doi:10.1080/27669645.2021.2015870**

Thank you for the suggestion. We have incorporated the recommended citation (doi:10.1080/27669645.2021.2015870) into our manuscript, along with additional relevant publications from the same research group to enrich our references.

- **L80: "49% of the total inflow into Qinghai Lake, approximately" to "~49% of the total inflow into Qinghai Lake"**

Thanks. The sentence was rewritten as you suggestion.

- **Figure 1 can be replaced by Figure 1, and the Figure 1 is not necessary as they are very similar.**

Thank you for your comment. If I understand correctly, you're suggesting that Figure 1 be replaced by Figure 5. However, we believe it's important to retain both figures. Figure 1 provides the terrestrial background and location of Qinghai Lake, while Figure 5 illustrates the domain decomposition created by the rSHUD tool. The triangular mesh in Figure 5 represents the computational domains where the algorithm is applied, offering readers a clear understanding of the model domains.

To differentiate between Figure 1 and Figure 5, we have added the following explanation:

"While Figure 5 might seem similar to Figure 1 at first glance, it specifically represents the unstructured triangular mesh model domains constructed by the rSHUD tool, highlighting the computational framework applied in our study."

- **L85: correct the type of citation "biogeochemistry Su et al. (2019, 2020)."**

Thanks. The typo is corrected in the revision.

- **L220: "located at 99○44'13"E, 37○02'13"N" can be moved to the study area.**

Thanks. The coordinates were removed as you suggestion.

- **L230, L345, L260 and others: "Figure 5 elucidates the domain", "Figures 6 and 7 show the model's". The writing can be improved by describing the results directly and then quoting the figures at the end of the sentence.**

Thank you. The two sentences was rewritten as following:

"The domain decomposition results for the Qinghai Lake Basin (QLB) are detailed in Figure 5. "

and

"The model's ability to simulate daily and monthly discharges in the Buha River is demonstrated in Figures 6 and 7, respectively."

- **Figure 8: The text size can be reduced. Please check all figures and use the consistent text size.**

Thank you for your feedback. We acknowledge the importance of consistent text size across all figures for clarity and readability. We will ensure consistent text size across all figures and will make necessary adjustments in consultation with the figural editor.

- **Conclusion: A sentence of lake water balance for Qinghai Lake from the evaluation of this model can be summarized.**

Thank you for your suggestion. We appreciate the emphasis on summarizing the lake water balance for Qinghai Lake from our model's evaluation. However, the primary objective of our manuscript is to introduce the model and demonstrate its potential in lake-watershed coupling research, rather than to provide a definitive conclusion on

Qinghai Lake's hydrology. Ongoing research on Qinghai Lake, conducted in collaboration with other groups, is more suited to draw comprehensive conclusions about the lake's hydrology. Therefore, we have focused our conclusion on the model's capabilities and applications, leaving detailed hydrological assessments of Qinghai Lake to future, more specialized studies.

---

## Author Response (AR2)

**Reply to Editor.**

Following are my reply to the editor's comments.

The editor's comments are in bold, and my reply is in blue and normal text.

**Please ensure that the colour schemes used in your maps and charts allow readers with colour vision deficiencies to correctly interpret your findings. Please check your figures using the Coblis – Color Blindness Simulator (https://www.color-blindness.com/coblis-color-blindness-simulator/) and revise the colour schemes accordingly (see F6). For the next revision, I also kindly ask you to remove the text part "Copyright statement. TEXT" on page 1.**

reply:

1)  We did the Color Blindness screening, and updated the figure 1, 8 and 9.

   • Figure 1: we added the annotation of Buha River and the the gauge station.

   • Figure 8(a): We updated the line types of the simulated and observed lake level, therefore this lines are more discernible.

   • Figure 9: The line width of lines in bottom subfigure are thicker than the previous version.

2)  The "Copyright statement. TEXT" was removed in the revision.